# Intracranial Assessment of Androgen Receptor Antagonists in Mice Bearing Human Glioblastoma Implants

**DOI:** 10.3390/ijms25010332

**Published:** 2023-12-26

**Authors:** Nomi Zalcman, Liraz Larush, Haim Ovadia, Hanna Charbit, Shlomo Magdassi, Iris Lavon

**Affiliations:** 1Leslie and Michael Gaffin Center for Neuro-Oncology, Hadassah Medical Center, Faculty of Medicine, The Hebrew University of Jerusalem, Jerusalem 91120, Israel; nomizlc@gmail.com (N.Z.);; 2Agnes Ginges Center for Human Neurogenetics, Department of Neurology, Hadassah Medical Center, Faculty of Medicine, The Hebrew University of Jerusalem, Jerusalem 91120, Israel; ovadiafam@gmail.com; 3Casali Center, Institute of Chemistry, The Hebrew University of Jerusalem, Jerusalem 91120, Israel; liraz.larush@gmail.com (L.L.); magdassi@mail.huji.ac.il (S.M.)

**Keywords:** glioblastoma, androgen receptor (AR), enzalutamide, bicalutamide, androgen receptor antagonists

## Abstract

The median survival time of patients with an aggressive brain tumor, glioblastoma, is still poor due to ineffective treatment. The discovery of androgen receptor (AR) expression in 56% of cases offers a potential breakthrough. AR antagonists, including bicalutamide and enzalutamide, induce dose-dependent cell death in glioblastoma and glioblastoma-initiating cell lines (GIC). Oral enzalutamide at 20 mg/kg reduces subcutaneous human glioblastoma xenografts by 72% (*p* = 0.0027). We aimed to further investigate the efficacy of AR antagonists in intracranial models of human glioblastoma. In U87MG intracranial models, nude mice administered Xtandi (enzalutamide) at 20 mg/kg and 50 mg/kg demonstrated a significant improvement in survival compared to the control group (*p* = 0.24 and *p* < 0.001, respectively), confirming a dose–response relationship. Additionally, we developed a newly reformulated version of bicalutamide, named “soluble bicalutamide (Bic-sol)”, with a remarkable 1000-fold increase in solubility. This reformulation significantly enhanced bicalutamide levels within brain tissue, reaching 176% of the control formulation’s area under the curve. In the U87MG intracranial model, both 2 mg/kg and 4 mg/kg of Bic-sol exhibited significant efficacy compared to the vehicle-treated group (*p* = 0.0177 and *p* = 0.00364, respectively). Furthermore, combination therapy with 8 mg/kg Bic-sol and Temozolomide (TMZ) demonstrated superior efficacy compared to either Bic-sol or TMZ as monotherapies (*p* = 0.00706 and *p* = 0.0184, respectively). In the ZH-161 GIC mouse model, the group treated with 8 mg/kg Bic-sol as monotherapy had a significantly longer lifespan than the groups treated with TMZ or the vehicle (*p* < 0.001). Our study demonstrated the efficacy of androgen receptor antagonists in extending the lifespan of mice with intracranial human glioblastoma, suggesting a promising approach to enhance patient outcomes in the fight against this challenging disease.

## 1. Introduction

Glioblastoma is a primary brain tumor and one of the most aggressive solid tumors in current oncological patients [1]. The documented median survival time is 15 to 19 months. The current standard of care includes safe surgical resection followed by radiotherapy and Temozolomide (TMZ) per os and adjuvant chemotherapy with TMZ. Currently, the chemotherapy regimens approved for glioblastoma, except the TMZ, are limited to carmustine and bevacizumab, which are primarily inefficient [2]. Therefore, we followed this unmet need, explored a new therapeutic target for this disease, and found that the androgen receptor (AR) is amplified at the DNA and RNA levels, and 56% of glioblastomas over-express AR protein [3]. We also demonstrated that AR protein expression could be detected in glial tumors in real time by positron emission tomography/computed tomography scanning using [F18] DHT [4].

AR functions as a steroid-hormone-activated transcription factor in the cytoplasm, bound to chaperone proteins in the heat-shock family. Upon binding of androgens like testosterone or dihydrotestosterone (DHT), the chaperones are released, allowing AR to homodimerize, translocate to the nucleus, and stimulate the transcription of AR-responsive genes, known for promoting cell proliferation and migration, primarily in prostate growth [5]. Based on its natural role, AR’s involvement in prostate cancer has been prominent and known for many years [6]. However, the involvement of AR in glioblastoma and the potential androgen receptor-regulated signaling pathways in glioblastoma were largely unknown.

The two most abounded AR antagonists are bicalutamide and enzalutamide. Bicalutamide is a second-generation AR antagonist designed to prevent androgens from binding to AR. Enzalutamide, a third-generation AR antagonist, inhibits AR nuclear translocation and AR binding to DNA [7]. Our observations of AR overexpression in glioblastoma and the known efficacy of AR inhibitors in prostate cancer prompted the investigation of AR antagonists’ impact on glioblastoma cells. This exploration revealed a significant (*p* < 0.05) concentration-dependent induction of cell death in three glioblastoma cell lines and two glioma-initiating cell lines upon treatment with AR antagonists. Furthermore, the administration of 20 mg/kg Xtandi (enzalutamide) via the oral route to nude mice hosting subcutaneous U87MG human glioblastoma xenografts yielded a remarkable 72% reduction in tumor volume (*p* = 0.0027) [3], consistent with Werner C.K. et al. [8].

Motivated by these results, the current study aimed to ascertain the therapeutic potential of AR antagonists in intracranial models of human glioblastoma.

## 2. Results

### 2.1. Xtandi^®^ Treatment at 20 mg/kg and 50 mg/kg Extends Lifespan in Intracranial U87MG Glioblastoma-Bearing Mice

Mice bearing intracranially implanted U87MG cells (human glioblastoma) and subjected to treatment with Xtandi^®^ (enzalutamide) at dosages of 20 mg/kg and 50 mg/kg demonstrated a significant extension in their lifespan compared to the control group that received the vehicle treatment (log-rank test; z = 2.26, *p* = 0.24 and z = 4.08, *p* < 0.001, respectively). Notably, a dose–response relationship is evident, as the cohort administered the 50 mg/kg Xtandi^®^ dosage exhibited a more pronounced lifespan enhancement than the 20 mg/kg dosage group (Figure 1). It is important to highlight that the results presented in Figure 1 for both the vehicle group and the 20 mg/kg Xtandi group are amalgamated from two separate experiments. The initial experiment comprised a vehicle-treated group (n = 9) and a group treated with 20 mg/kg Xtandi (n = 5) (see Appendix A). Subsequently, an additional study was conducted, involving three study groups: vehicle (n = 11), 20 mg/kg Xtandi (n = 7), and 50 mg/kg Xtandi (n = 10) (refer to Appendix A). In light of the absence of discernible differences between the vehicle groups or the 20 mg/kg groups in the two studies, we opted to merge them for a more consolidated representation (Figure 1).

To fortify our findings and evaluate the efficacy of an alternative androgen receptor antagonist, bicalutamide was reformulated using excipients recognized as GRAS, as detailed in the Section 4. The reformulation yielded unexpected results: The lipophilic active agent, bicalutamide, when included in the formulation prepared per the disclosed process, exhibited a remarkable increase in water solubility. The solubility increased from 0.005 mg/mL to 1–5 mg/mL at 20 °C, constituting a minimum 1000-fold solubility enhancement. In light of this result, this bicalutamide-soluble formulation earned the moniker “Bic-sol”. Notably, the Bic-sol solution remained transparent for several consecutive days.

### 2.2. Bicalutamide Formulation Yields Clear Solution

Cryo-TEM analysis revealed that the bicalutamide formulation, reconstituted at either 1% or 5% *w*/*w* of the total powder in water, was substantially devoid of particles, with a few liposome-like shapes apparent at the largest magnification (200 nm scale), similar to the negative control formulation (Figure 2). The size of the liposome-like structures that were sparsely identified in the bicalutamide-containing formulation and the formulation lacking the active ingredient was about 100–120 nm. Further analysis using dynamic light scattering (DLS) revealed no particles in the bicalutamide-containing or negative control formulation.

### 2.3. Bicalutamide Active Ingredient Presence in Aqueous Phase

Analysis of the new formulation using HPLC liquid chromatography and tandem mass spectrometry (HPLC-MS/MS) demonstrated a distinctive peak at *m*/*z* 429. This peak is characteristic of bicalutamide, which has a molecular weight (MW) of 430 g/mol. In negative ionization mode ((Mw-H) = 429 g/mol), this peak was only present in the bicalutamide sample and was not detected in the control sample. This unequivocally confirmed the presence of bicalutamide in the filtrate (Figure 3A). Additionally, when the mass spectrometer was set to negative ionization mode with deprotonated ions (M-H), the recorded transition corresponded to values that have been previously associated with bicalutamide (*m*/*z* 429.2 → 255.0) [9] (Figure 3B).

In summary, the results indicate the successful formulation of bicalutamide, yielding a clear solution upon reconstitution in water that remained stable for at least several days. Visual inspection, dynamic light scattering (DLS), and cryo-TEM analysis of formulations dispersed at 1% or 5% *w*/*w* revealed the absence of particles.

### 2.4. The New Bicalutamide Formulation, Bic-sol, Demonstrates Improved Pharmacokinetics

The plasma pharmacokinetics (PK) with brain penetration of bicalutamide following a single intravenous (IV) or oral administrations of the new bicalutamide formulation were tested in male CD1 mice and compared to the control bicalutamide formulation. The PK parameters are presented in Table 1. Mean plasma and brain concentration–time profiles of bicalutamide after a single dose (N = 15/group) are depicted in Figure 4A and Figure 4B, respectively.

Following an intravenous dose of the bicalutamide test formulation, the concentration of bicalutamide in mouse plasma exhibited a terminal half-life (T_1/2_) of 28.5 h. The area under the curves from time 0 to the last time point (AUC_last_) and from time 0 to infinity (AUCINF) were 188,799 and 206,498 h/ng/mL, respectively. The total clearance and volume of distribution at steady state (Vss) were 0.0387 L/h/kg and 1.42 L/kg, respectively. In mouse brains, the concentration of bicalutamide declined with a half-life of 21.2 h, reaching a C_max_ value of 1730 ng/mL with a mean Tmax of 4.00 h. The AUC_last_ and AUCINF in the brain were 80,850 h ng/mL and 83,499 h ng/mL, respectively. Notably, the concentration ratio of brain to plasma at various time points post-IV administration was measured as 0.351, 0.380, 0.307, 0.306, and 0.206 folds (Table 1 and Figure 4A).

For the control bicalutamide formulation administered orally, plasma concentration declined with a half-life of 28.3 h, reaching a C_max_ of 3610 ng/mL with a mean T_max_ of 8.00 h. The AUC_last_ and AUCINF in plasma were 162,774 and 180,808 h ng/mL, respectively. The oral bioavailability (F) was calculated at 87.6% (Table 1 and Figure 4A). In mouse brains, the concentration of bicalutamide declined with a half-life of 23.6 h, reaching a C_max_ of 829 ng/mL with a mean T_max_ of 8.00 h. The AUC_last_ and AUCINF in brain tissue were 39,281 h ng/mL and 41,341 h ng/mL, respectively (Table 1 and Figure 4B). The concentration ratio of brain to plasma at various time points post-PO administration of the control bicalutamide formulation was measured as 0.240, 0.308, 0.230, 0.202, and 0.138 folds.

These results demonstrate that the new bicalutamide formulation exhibits improved pharmacokinetic properties compared to the control formulation. Notably, the mean AUC_last_ of the test formulation following PO administration was 132% of that obtained by the control formulation, surpassing the enhancement measured following IV administration, where the AUC_last_ of the test formulation was 116% of the control formulation. Surprisingly, the enhancement in bicalutamide levels was even greater in brain tissue than in plasma, with brain tissue levels reaching 176% of those in the control formulation, compared to 142% in plasma.

### 2.5. The Newly Formulated Bic-sol Significantly Extends Mouse Survival in the U87MG and the GIC-ZH161 Intracranial Model

In the U87MG intracranial mice model, 2 mg/kg and 4 mg/kg of the newly formulated Bic-sol demonstrated significant efficacy compared to mice treated with the vehicle, as evidenced by differences in hazard rates. Specifically, the hazard rates were notably distinct between the vehicle and 2 mg/kg Bic-sol (log-rank test; z = 2.37, *p* = 0.0177) as well as between the vehicle and 4 mg/kg Bic-sol (log-rank test; z = 2.91, *p* = 0.00364). However, there was no significant difference observed between the administration of 2 mg/kg and 4 mg/kg of solubilized bicalutamide (referred to as “Bic-sol”) when analyzed using the (log-rank test z = 0.81, *p* = 0.42) (Figure 5A). Additionally, combination therapy of 8 mg/kg Bic-sol with TMZ (Temozolomide) demonstrated superior efficacy compared to 8 mg/kg Bic-sol as a monotherapy (log-rank test; z = 2.69, *p* = 0.00706), as well as compared to TMZ as a monotherapy (log-rank test; z = 2.36, *p* = 0.0184). Monotherapy of 8 mg/kg of Bic-sol administration resulted in substantial differences in hazard rates (log-rank test; z = 4.7, *p* < 0.001) compared to mice treated with the vehicle (Figure 5B). It is worth noting that the control bicalutamide formulation failed to extend the lifespan of the mice (Figure 5A). 

In the ZH-161 mice model, the group’s lifespan treated with 8 mg/kg Bic-sol as monotherapy was significantly longer than the group treated with TMZ (log-rank test; z = 4.28, *p* < 0.001) or with the vehicle (log-rank test; z = 4.69, *p* < 0.001) (Figure 6).

## 3. Discussion

The findings of this study align with and extend the growing body of literature that highlights the pivotal role of androgen receptor (AR) antagonists in the treatment of glioblastoma. Glioblastoma, characterized by its aggressive nature and limited treatment options, has long been a formidable challenge in the field of oncology [1,2]. A noteworthy development is the recognition of AR expression in a considerable percentage of glioblastoma cases, suggesting a promising avenue for potential therapeutic interventions [3,10,11,12,13,14,15,16,17,18,19].

Our study’s demonstration of the efficacy of enzalutamide and a newly formulated bicalutamide in extending the lifespan of mice bearing intracranial human glioblastoma is consistent with earlier in vitro and in vivo studies. Notably, both our group and others have conducted studies involving enzalutamide, which have shown a significant reduction in glioblastoma tumor volume in mouse subcutaneous xenograft models, echoing these findings [8].

As all AR antagonists were initially developed for prostate cancer, there remains an unmet medical need for additional therapeutic modalities for brain tumors, such as glioblastoma, and a demand for improved formulations of AR antagonists specifically tailored for brain tumors to enhance bioavailability. Bicalutamide is well absorbed after oral administration due to its high lipophilicity (log P, 2.92); however, it exhibits very low water solubility (<40 mg/L) [20]. Thus, bicalutamide poses challenges for effective oral administration due to its low bioavailability. Various strategies, including solid dispersions [21], micellar solubilization [22], nanoemulsions [23], hydrotropic solubilization [24], particle size reduction [25], lipid-based delivery systems [26], and complexation with cyclodextrins (CyDs) [27], are employed to enhance the water solubility of lipophilic drugs. We addressed this need through the reformulation of bicalutamide with excipients known as “GRAS” and the development of novel drug formulations, “Bic-sol” introduced in this study. Our research shows that this formulation dramatically enhances the solubility of bicalutamide, leading to increased concentrations in brain tissue and improved bioavailability. In contrast to other solubility methods employed for enhancing the water solubility of lipophilic drugs, the approach utilized in this study—dissolving the active ingredient in acetone—offers distinct advantages. The choice to utilize a polar solvent like acetone was based on the anticipation that a clear solution would form when mixed with water and another polar solvent (isopropanol) containing surfactants. The key differentiator lies in achieving complete molecular-level dissolution of the active ingredient at this stage, as opposed to existing in the form of particles or droplets. The choice to utilize a polar solvent like acetone was based on the anticipation that a clear solution would form when mixed with water and another containing surfactants. The key differentiator lies in achieving complete molecular-level dissolution of the active ingredient at this stage, as opposed to existing in the form of particles or droplets. Comparatively, alternative methods, such as those relying on different solvents or co-solvents, may yield dispersed powders or encounter challenges in achieving the desired molecular-level solubility. Some methods might introduce potential issues related to the formation of particles—either too small and toxic or too large and hindered in penetrating biological barriers. The significance of our approach becomes apparent in the resulting water-soluble powder after the drying process, which demonstrates complete dissolution in water. This stands in contrast to methods that might produce powders with varying degrees of solubility or stability issues related to particle aggregation.

In summary, the chosen method prioritizes achieving molecular-level solubility, thus offering advantages over other techniques that may face challenges in controlling particle size, ensuring complete dissolution and addressing potential toxicity or stability concerns. Similar investigations [28,29,30] have explored various solubilization techniques for AR antagonists, emphasizing the need for improved drug delivery systems to maximize their therapeutic potential. These studies, in conjunction with our findings, underscore the importance of enhancing drug solubility as a key strategy in neoplasm treatment development.

In the U87MG intracranial mouse model, Bic-sol monotherapy at 2 mg/kg and 4 mg/kg doses demonstrated significant efficacy, with distinct hazard rate differences compared to the control group. Combining Bic-sol with TMZ showed superior efficacy compared to monotherapy, highlighting its potential as a valuable addition to glioblastoma treatment. Importantly, Bic-sol monotherapy significantly extended mouse survival when compared to the control bicalutamide formulation, likely due to its improved brain bioavailability. These findings were consistent in the ZH-161 mouse model, with Bic-sol monotherapy proving notably more effective in extending the mouse lifespan compared to TMZ or the control group, emphasizing its therapeutic potential.

It is important to acknowledge our prior publication, where we identified the expression of the AR variant (AR-V7) in ZH-161, potentially linked to androgen antagonist resistance in prostate cancer [3]. The juxtaposition of this finding with our earlier revelation of only a moderate response of ZH-161 to androgen receptor antagonists in in vitro studies [3] introduces an intriguing aspect. The efficacy of Bic-sol in the in vivo ZH-161 model prompts the consideration of potential explanations for this apparent discrepancy. Firstly, the resistance observed in prostate cancer may not seamlessly translate to glioblastomas, where distinct factors could influence the drug’s efficacy, potentially accounting for the observed sensitivity in the ZH-161 model. Additionally, despite the expression of AR-V7, the ZH-161 line co-expresses AR-FL, adding an extra layer of complexity to the observed responses. The concurrent presence of AR-FL might contribute to the efficacy gained by the drug. This observation could strengthen our results. If the drugs exhibit efficacy in a cell line with AR-V7, coupled with the fact that only 30% of glioblastoma patients have AR-V7, it raises the intriguing possibility that these drugs might exert even greater efficacy in patients lacking this specific transition. While we acknowledge that these assertions are drawn from our observations and represent assumptions, future investigations, supported by rigorous clinical studies, will be imperative to validate and build upon these preliminary findings, deepening our understanding of the intricate interplay between AR-V7, AR-FL, and drug efficacy in the context of glioblastoma.

Notably, this new formulation of Bic-sol offers the additional benefit of reducing the amount of bicalutamide [31], potentially mitigating side effects associated with its use. This improvement in formulation underscores the potential for enhanced treatment outcomes with a more favorable safety profile. Furthermore, when compared to enzalutamide, it is noteworthy that Bic-sol demonstrated comparable efficacy at 10-times lower doses, emphasizing its efficiency in glioblastoma treatment. While we highlight the improved characteristics of Bic-sol, including enhanced solubility and increased bioavailability, comprehensive safety assessments are imperative, especially in combination with standard-of-care chemotherapy (e.g., TMZ). Future research should delve into the potential adverse effects and long-term safety implications associated with the use of AR antagonists.

The clinical translation of our research findings into a PET/CT-guided human clinical trial is a significant step forward, aligning with the vision of precision medicine. The utilization of the fluoro-5α dihydrotestosterone tracer for identifying high AR expression in glioblastoma tumors, as pioneered by Orevi et al. [4], is a notable advancement. The work of Orevi et al. highlights the feasibility of certifying patients for AR antagonist therapy based on the presence of AR protein in their tumors, thereby opening doors for personalized treatment approaches. This aligns with our pursuit of personalized therapy based on the level of AR expression within glioblastoma tumors, bringing us closer to tailoring treatments to individual patients.

However, the journey from promising preclinical results to effective clinical therapy is complex [32,33]. The efficacy of AR antagonists, demonstrated in our mouse models, may not universally apply to all glioblastoma cases. Tumor heterogeneity and patient-specific factors such as MGMT promoter methylation status should be considered when assessing the potential generalization of our findings, including the emergence of resistance mechanisms, intricacies in combining multiple therapeutic agents, and the potential for side effects. Furthermore, ethical considerations, regulatory approvals, cost factors, accessibility, and patient preferences will significantly influence the translation of our research findings into a clinically viable treatment option [32,33,34]. Gan X. et al.’s analysis underscores the importance of addressing these obstacles to ensure the ultimate clinical effectiveness of AR antagonist therapy in glioblastoma patients [11]. It is crucial to emphasize that, in controlled clinical studies, 0.5% (10 out of 2051) of patients experienced seizures following exposure to enzalutamide. Notably, the incidence of seizures seems comparable among patients with metastatic prostate cancer, sharing similar seizure risk factors, whether exposed to enzalutamide or not. This suggests that enzalutamide may offer benefits for patients with a history of seizures or other predisposing factors [31]. However, given the absence of exploration of this treatment in patients with brain tumors, it is imperative to closely monitor each patient throughout the duration of their treatment.

In conclusion, our study contributes to the expanding body of literature supporting AR antagonists and innovative drug formulations as potential game-changers in glioblastoma treatment. The promise they hold, coupled with personalized approaches based on AR expression, provides hope for improved therapeutic strategies in the battle against this challenging disease. However, it is clear that further interdisciplinary efforts are essential to navigate the intricate path from laboratory research to clinical success, ultimately benefiting glioblastoma patients and their families.

## 4. Materials and Methods

### 4.1. Cell Culture

U87MG was obtained from the American Type Culture Collection (Manassas, VA, USA). They were cultured in Dulbecco’s modified Eagle’s medium (DMEM) supplemented with 4 mmol/L L-glutamine, 100 units/mL penicillin, 100 μg/mL streptomycin, and with (as indicated) 10% of FBS. The glioma-initiating cell (GICs) lines, ZH-161, were kindly provided by Prof. Michael Weller from the Department of Neurology at the University Hospital Zurich, Switzerland, and maintained as described [35,36]. Briefly, cells were cultured in Neurobasal Medium (Gibco; Thermo Fisher Scientific, Inc., Waltham, MA, USA) supplemented with B-27 (20 μL/mL) and glutamax (10 μL/mL), fibroblast growth factor (FGF)-2, epidermal growth factor (EGF) (20 ng/mL each (Peprotech, Rocky Hill, PA, USA), and heparin (32 IE/mL; Sigma-Aldrich, St. Louis, MO, USA). All cells were maintained in a humidified incubator at 37 °C in 5% CO_2_.

### 4.2. In Vivo Inhibition of Glioblastoma Growth

The potential of androgen receptor antagonists to inhibit human glioblastoma growth was studied in an intracranial in vivo model.

Ethical statement: This study was carried out in accordance with the recommendations in the Guide for the Care and Use of Laboratory Animals of the National Institutes of Health. The protocol was approved by the Committee on the Ethics of Animal Experiments of the Hebrew University Medical School (Permit Number: MD-16-14864-5 and MD-18-15591-5). To minimize the suffering of the animals, injections of tumor cells were performed under light anesthesia (ketamine + xylazine, 100 and 5 mg/kg body weight, respectively). The animals were monitored twice weekly for body weight and neurological signs. Upon termination of the experiments, the animals were euthanized by exposure to excess CO_2_.

In this study, 6- to 8-week-old athymic male nude (nu/nu) mice were intracranially implanted with U87MG glioblastoma cells (4 × 10^5^ cells) or spheres of ZH-161 glioblastoma-initiating cells (GICs) (1 × 10^5^ cells) into their right cerebral hemisphere (1 mm posterior and 2.3 mm lateral to the bregma, to a depth of 3 mm). After 7 days, once tumors had been established, tumor-bearing mice were randomized into the treatment groups (n = 5–10 per group), as indicated. Each group was treated 5 times weekly by oral gavage with either vehicle (vehicle of Xtandi was composed of 220 mg/kg caprylocaproyl polyoxylglycerides in saline and vehicle of Bic-sol contained empty powder soluble in DW); 20, 50, or 100 mg/kg of a commercially available enzalutamide formulation (Xtandi, Astellas Pharma Inc., Tokyo, Japan); bicalutamide (Sigma-Aldrich) dissolved in 5% DMSO and 95% corn oil; the newly formulated bicalutamide (Bic-sol) prepared as described below; or with TMZ (0.75 mg/kg 3% DMSO in saline) (Tocris Bioscience, Bristol, UK). The endpoint of the experiment was defined as the number of days that elapsed from tumor implantation to the day of overt symptoms (significant weight loss, lethargy, hunched posture, or other neurological signs).

### 4.3. Preparation of an Improved AR Antagonist Formulation

The process for producing the test formulations comprising AR antagonists is as follows: 0.225 g of Lecithin S75 (LIPOID^®^ S75, Lipoid, Newark, NJ, USA), 0.225 g of ammonium glycyrrhizinate (Sigma), 9.5 g of isopropanol (Alfaaesar), and 5.5 g of TDW (Triple Distilled Water) were combined in a glass vial and mixed by magnetic stirrer until a clear solution was obtained. Then, 0.05 g of the active drug (Bicalutamide, Apex Biotech LLC (Boston, MA, USA), or enzalutamide, A2S (Yavne, Israel)) was dissolved in 1.5 g acetone. The two solutions were combined and mixed with a magnetic stirrer until a clear solution was obtained. The solution was spray dried (Mini Spray Dryer B-290) (buchi Sarl, Villebon sur Yvette, France) at an air inlet temperature of 120 °C and a liquid feed rate of 10 mL·min^−1^. The powder obtained at the end of the spray-dry process was dispersed, 1% *w*/*w*, in distilled water by vortex for 5 min.

### 4.4. Characterization of the New Formulation

#### 4.4.1. Cryo-TEM Analysis

The formulation products were analyzed using transmission electron cryomicroscopy (Cryo-TEM). The formulations comprising active ingredients were compared to those without any active material. The amount of the active ingredient (bicalutamide) in the powder before dispersion was 10% *w*/*w*. To prepare samples for analysis, 1% or 5% (as indicated) of the total weight of the powder was added to a vial with water (TDW) and vortexed for 5 min. Then, 2–4 µL drop of the test sample was applied to a TEM grid (300 mesh Cu Lacey substrate, Ted Pella, Ltd., Redding, CA, USA) following a short pre-treatment of the grid by glow discharge. The excess liquid was blotted off, and the specimen was vitrified by rapid plunging into liquid ethane pre-cooled by liquid nitrogen using a vitrification robot system (Vitrobot mark IV, FEI) (Thermo Fisher Scientific Inc., Waltham, MA, USA).

#### 4.4.2. Particle Size Distribution Measurements by Dynamic Light Scattering (DLS)

Particle sizes after re-dispersion of the obtained powders (with and without bicalu-tamide) in water were measured at room temperature by DLS using a Nano-ZS Zetasiz-er (Malvern Instrument Ltd., Worcestershire, UK). The instrument is equipped with 633 nm laser, and the light scattering is detected at 173 degrees by backscattering technology (NIBS, Non-Invasive Backscatter). The re-dispersed powders in water (1% *w*/*w* of powder in water) were performed three times, and each measurement was performed in triplicate.

#### 4.4.3. HPLC Liquid Chromatography and Tandem Mass Spectrometry (HPLC-MS/MS)

To examine whether the bicalutamide active ingredient is present in the aqueous phase of the reconstituted formulation, samples were subjected to microfiltration, and high-performance liquid chromatography (HPLC) analysis was performed on the aqueous filtrate. To this end, formulations prepared as described above, with or without bicalutamide, were reconstituted in water (1% *w*/*w*) and filtered through a 0.22-micron Millipore filter. Then, 100 µL of the filtrate was injected into the HPLC system.

The presence of bicalutamide in the aqueous filtrate was further confirmed by liquid chromatography and tandem mass spectrometry (HPLC-MS/MS), essentially as described by Kim et al. [9]. The mass spectrometer was tuned in negative ionization mode with deprotonated ions (M-H).

#### 4.4.4. Pharmacokinetic Analyses of Bic-sol

The plasma pharmacokinetics (PK) with brain penetration of bicalutamide following a single intravenous [7] or oral (PO) administration of the new bicalutamide formulation was tested in male CD1 mice and compared to that of the control bicalutamide formulation.

For the preparation of the new bicalutamide formulation for IV/PO dosing in mice, a bicalutamide formulation was prepared, as described in Example 1, combined with water to obtain a concentration of 20 mg/mL of the bicalutamide formulation in DDW, and this was vortexed for two minutes. The appearance of the resulting test formulation (herein designated “new bicalutamide formulation” or “bicalutamide test formulation”) was of a colorless transparent solution.

To prepare the control bicalutamide formulation for PO dosing for mice, bicalutamide was dissolved to obtain a concentration of 2 mg/mL bicalutamide in 5% DMSO and 95% corn oil as follows. Bicalutamide was first dissolved in DMSO, combined with corn oil, and vortexed for two minutes. The appearance of the resulting control formulation (herein designated “control bicalutamide formulation” or “bicalutamide control”) was of a clear yellow solution.

The concentration of the bicalutamide active ingredient in both the resulting test formulation and control formulation was 2 mg/mL.

A total of 45 male CD1 mice, approximately 24~25 g of body weight, were purchased from JH Laboratory Animal Co., Ltd (Seoul, Republic of Korea). The animals had free access to food and water. The IV dosing was conducted via the tail vein, and the PO dosing was conducted via oral gavage. The animals were anesthetized by carbon dioxide inhalation, and blood samples (~110 µL) were collected via facial vein puncture or cardiac puncture into K2EDTA tubes. Blood samples were put on ice and centrifuged at 4 °C and 2000× *g* for 5 min to obtain plasma samples within 15 min.

After blood collection, the animals were euthanized by carbon dioxide inhalation; a mid-line incision was made in the scalp, and the skin retracted. The skull overlying the brain was removed, and whole brain samples were collected, rinsed with cold saline, dried on filtrate paper, weighted, and snap-frozen by placing them into dry ice. Tissue samples were homogenized with homogenizing solution (PBS) before analysis.

Three treatment groups (N = 15) were tested: new bicalutamide formulation 80 mg/Kg, IV or PO administration, and control bicalutamide formulation (8 mg/kg), PO administration (the dose of the active ingredient administered in all treatment groups was 8 mg/kg). Sampling was performed at 0.5, 1, 2, 4, 8, 12, 24, 48, and 96 h after dosing (nine time points, semi-serial bleeding for plasma, N = 3/time point). Terminal collection of brain samples was performed at 2, 4, 8, 12, and 96 h.

Blood samples were processed into plasma and analyzed for bicalutamide using a qualified liquid chromatography–tandem mass spectrometry (LC-MS/MS) method. Brain samples were analyzed for bicalutamide using a qualified liquid chromatography–tandem mass spectrometry (LC-MS/MS) method.

LC-MS/MS analysis was performed with the following parameters: instrument LC-MS/MS-19 (Triple Quad 5500) (SCIEX, Toronto, ON, Canada); Matrix Male CD1 mouse plasma and brain homogenate; analyte(s) bicalutamide; internal stand glipizide; MS conditions negative ion, ESI; SRM detection; compound name Parent (*m*/*z*)/Daught (*m*/*z*) bicalutamide; Q1/Q3 masses: 429.20/255.10 Da; glipizide Q1/Q3 masses: 444.30/319.10 Da; HPLC conditions Mobile phase: Mobile Phase A: H_2_O—0.025% FA—1 mM NH_4_OAC; Mobile Phase B: MeOH—0.025% FA—1 mM NH_4_OAC.
TimeParameter0.2050.60951.30951.3151.80Stop

Column: ACQUITY UPLC BEH C18 2.1 × 50 mm, 1.7 µm; column temperature: 60 °C; retention time: bicalutamide: 1.21 min; glipizide: 1.20 min.

Sample preparation: For non-diluted plasma samples: 1. An aliquot of 30 µL sample was added with 200 µL IS (Glipizide, 40 ng/mL) in ACN. 2. The mixture was vortexed for 1 min and centrifuged at 5800 rpm for 10 min. 3. A 100 µL supernatant was transferred to a new plate. 4. A solvent of 0.5 µL was injected into LC-MS/MS for analysis.

For brain samples:

The sample was homogenized with 3 volumes (*v*/*w*) of PBS. The diluted factor was 4. The following operation was the same as the undiluted ones.

The calibration curve was 10–10,000 ng/mL for bicalutamide in male CD1 mouse plasma and brain homogenate.

The PK parameters were determined based on group means by non-compartmental analysis using Pharsight Phoenix WinNonlin^®^ 8.2 software. “BQL” rule: concentration data under LLOQ (LLOQ = 10.00 ng/mL for bicalutamide in mouse plasma and brain homogenate) were replaced with “BQL” and excluded from graphing and PK parameter estimation. Terminal t½ calculation: time points were automatically selected by the “best fit” model for terminal half-life estimation as the first option. The manual selection was applied when “best fit” could not define the terminal phase well. When the number of detected time points after Tmax was less than 3, the terminal half-life and AUCINF were not calculated.

### 4.5. Statistical Analysis

Survival analysis, specifically employing Kaplan–Meier survival curves, was conducted to assess and compare survival rates among different treatment groups. Log-rank tests were utilized to identify statistically significant differences in survival distributions, and hazard ratios were calculated to quantify the risk of an event occurring in one group compared to another. All analyses were performed using the Evan Miller online tool (https://www.evanmiller.org/ab-testing/survival-curves.html, last accessed on 26 November 2023).

Pharmacokinetics data analysis, including parameters, such as CL (clearance), Vss (volume of distribution at steady state), T_max_ (time to maximum concentration), C_max_ (maximum concentration), T_1/2_ (half-life), AUC (area under the curve), F% (bioavailability), etc., was conducted using the non-compartmental model in WinNonlin V 8.2 statistical software (Pharsight Corporation, Sunnyval, CA, USA).

## Figures and Tables

**Figure 1 ijms-25-00332-f001:**
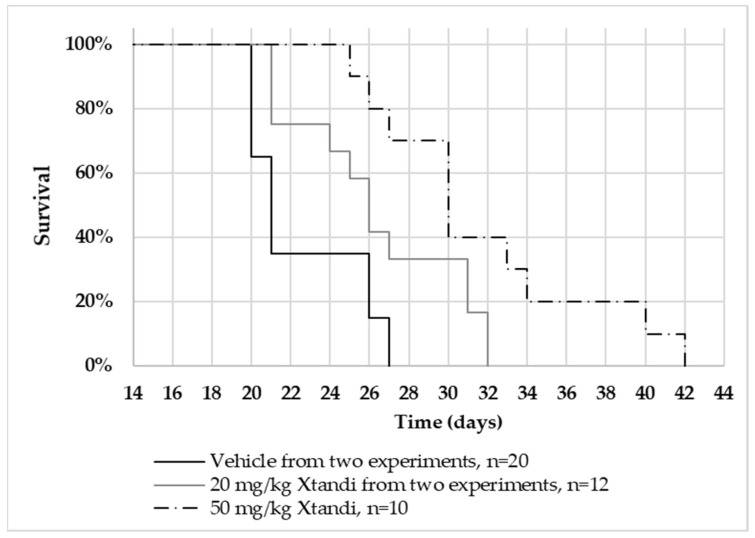
Kaplan–Meier analysis of nude mice with intracranial U87MG human glioblastoma tumors treated with enzalutamide (Xtandi). Nude mice with intracranial U87MG glioblastoma tumors were treated 5 times weekly with Xtandi or vehicle control. The survival of three groups was assessed: vehicle-treated mice (n = 20; solid black line, pooled from two experiments), mice treated with 20 mg/kg Xtandi (n = 12; solid grey line, also pooled from two experiments), and mice treated with 50 mg/kg Xtandi (n = 10; dash-dotted line).

**Figure 2 ijms-25-00332-f002:**
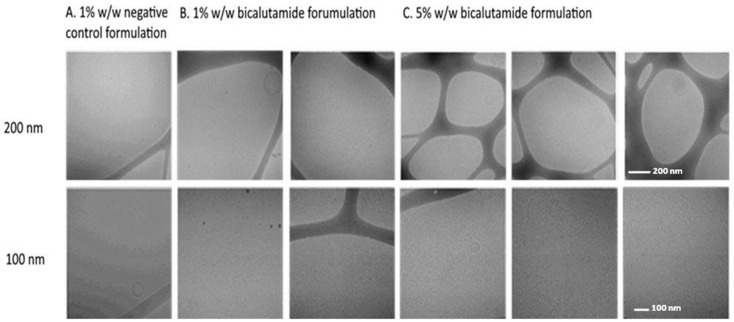
Cryo-TEM analysis of formulations. (**A**). Powder without the active drug dispersed in water (1% *w*/*w*, negative control formulation), (**B**) powder comprising bicalutamide dispersed in water (1% *w*/*w* bicalutamide formulation), and (**C**) powder comprising bicalutamide dispersed in water (5% *w*/*w* bicalutamide formulation). Scale bars represent 200 nm (**top** panels) or 100 nm (**bottom** panels).

**Figure 3 ijms-25-00332-f003:**
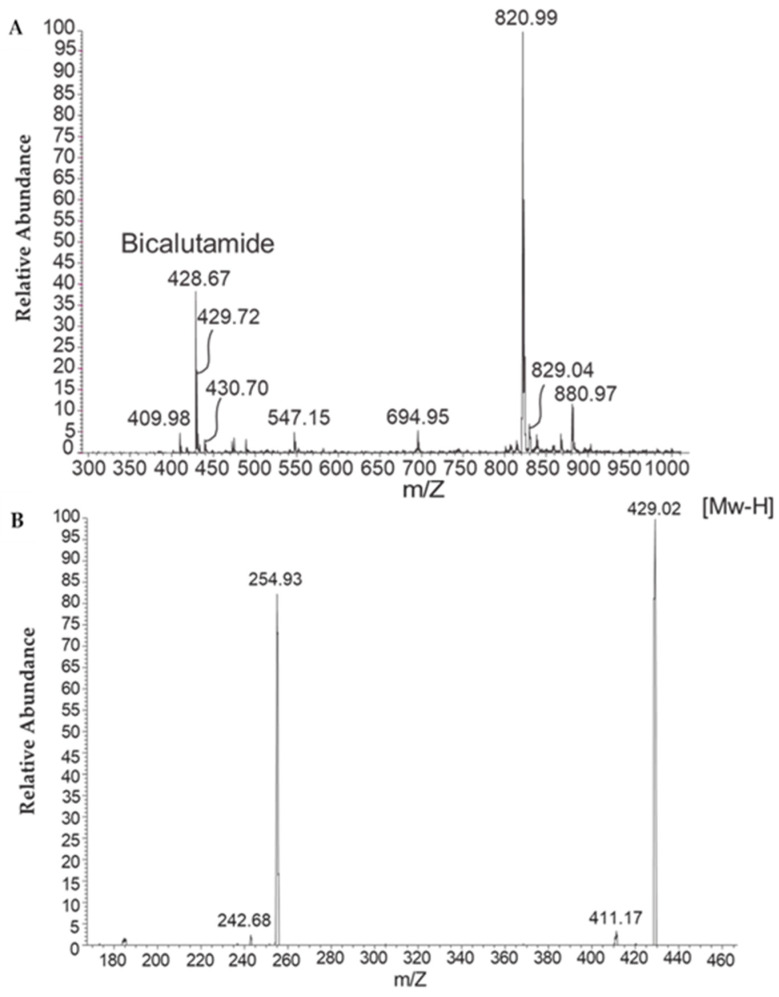
MS-HPLC chromatograms of filtrates of test formulations dispersed in water (1% *w*/*w*). (**A**) Formulation comprising bicalutamide, exhibiting a peak at 429 *m*/*z*, indicative of the presence of bicalutamide. (**B**) When the mass spectrometer was set to negative ionization mode with deprotonated ions (M-H), the recorded transition corresponded to values previously associated with bicalutamide (*m*/*z* 429.2 → 255.0).

**Figure 4 ijms-25-00332-f004:**
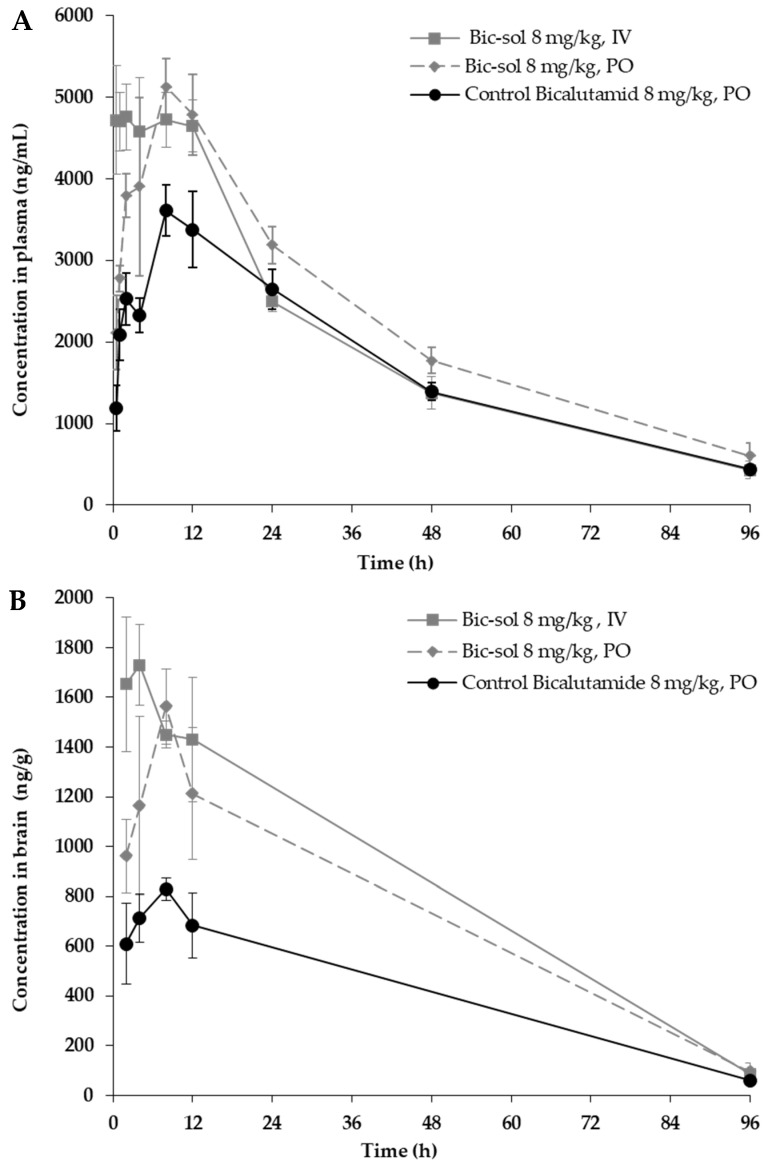
Mean concentration-time profiles of bicalutamide after single IV or PO dose to male CD1 mouse (N = 15/group). The new bicalutamide formulation (Bic-sol) was administered either intravenously [7] or orally (PO), with comparisons made to a control bicalutamide formulation administered orally. (**A**) Plasma concentration/time profiles; (**B**) brain concentration/time profiles. Study groups included: Bic-sol, IV administration (grey, solid line); Bic-sol, PO administration (grey dashed lines); control bicalutamide formulation, PO administration (Black solid line).

**Figure 5 ijms-25-00332-f005:**
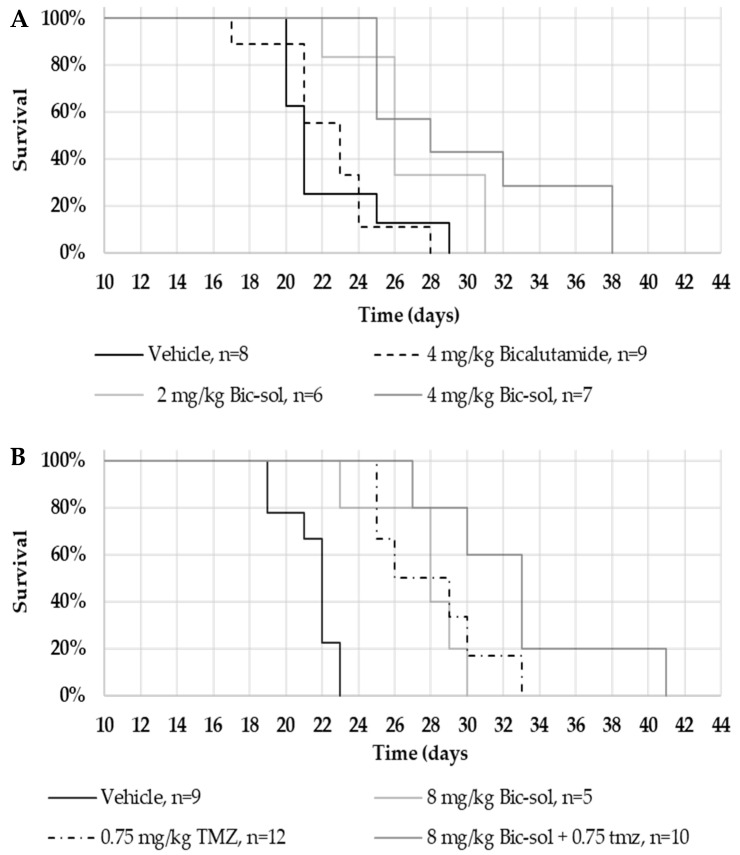
Kaplan–Meier analysis on nude mice implanted intracranially with U87MG human glioblastoma and treated with bicalutamide soluble (Bic-sol) and TMZ. U87MG cells were injected intracranially, and mice were treated 5 times weekly as indicated. (**A**) vehicle (n = 8; solid black line), 4 mg/kg bicalutamide dissolved in 5% DMSO in corn oil (n = 9; dashed line), 2 mg/kg (n = 6; light gray) or 4 mg/kg (n = 7; dark grey) Bic-sol dissolved in water. (**B**) vehicle (n = 9; solid black line), 8 mg/kg Bic-sol (n = 5; light gray line), 0.75 mg/kg TMZ (n = 12; dotted dashed line) or 8 mg/kg Bic-sol plus 0.75 mg/kg TMZ (n = 10; dark grey line). The animals were monitored for survival.

**Figure 6 ijms-25-00332-f006:**
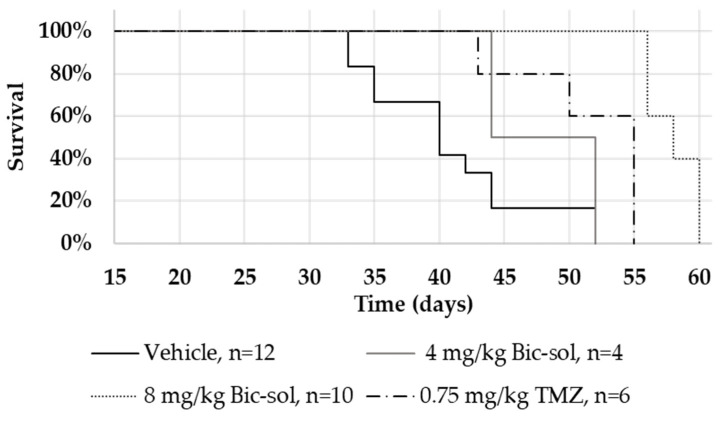
Kaplan–Meier analysis on nude mice implanted intracranially with GIC cells (ZH-161) and treated with bicalutamide soluble (Bic-sol) and TMZ. Vehicle (n = 12; solid black line), 4 mg/kg Bic-sol (n = 4; solid dark gray line), 8 mg/kg Bic-sol (n = 10; dotted line), 0.75 mg/kg TMZ (n = 6; dotted dashed line).

**Table 1 ijms-25-00332-t001:** PK parameters.

**Plasma IV—Bicalutamide Test Formulation**				**Brain IV—Bicalutamide Test Formulation**			
**CL, L/h/kg**	**Vss, L/kg**	**AUC_last_, ng h/mL**	**T_1/2_, h**	**T_max_, h**	**C_max_, ng/mL**	**AUC_last_, ng h/mL**	**T_1/2_, h**
0.0387	1.42	188,799	28.5	4.00	1730	80,850	21.2
**Plasma PO—Bicalutamide Test Formulation**				**Brain PO—Bicalutamide Test Formulation**			
**T_max_, h**	**C_max_, ng/mL**	**AUC_last_, ng h/mL**	**T_1/2_, h**	**T_max_, h**	**C_max_, ng/mL**	**AUC_last_, ng h/mL**	**T_1/2_, h**
8.00	5130	214,942	30.1	8.00	1563	69,060	22.3
**Plasma PO—Bicalutamide Control**				**Brain PO—Bicalutamide Control**			
**T_max_, h**	**C_max_, ng/mL**	**AUC_last_, ng h/mL**	**T_1/2_, h**	**T_max_, h**	**C_max_, ng/mL**	**AUC_last_, ng h/mL**	**T_1/2_, h**
8.00	3610	162,774	28.3	8.00	829	39,281	23.6

Abbreviations: CL (clearance), Vss (volume of distribution at steady state), T_max_ (time to maximum concentration), C_max_ (maximum concentration), T_1/2_ (half-life), AUC (area under the curve), F% (bioavailability).

## Data Availability

Data are contained within the article and Appendix A.

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
