# Peer review of "Intracranial Assessment of Androgen Receptor Antagonists in Mice Bearing Human Glioblastoma Implants"

_ijms, 2023, doi:10.3390/ijms25010332_

Round 1

Reviewer 1 Report

Comments and Suggestions for Authors

this study by Zalcman et al is a continuation of their previous studies on the role of AR in Glioblastoma and on the development of new antagonists aimed at inhibiting AR-induced intracellular signaling pathway in GBM.

the study is well-conducted and well-described

in my opinion, the manuscript can be accepted in this way but I suggest discussing the possibility of drug resistance due to the presence of alternative splicing forms of AR (see AR-V7 the authors previously described in GBM)

Comments on the Quality of English Language

English language is fine and fluent.

minor editing is required.  i.e:

1) line 99: "bic-sol" or "sol-bic"?

2) line 201 TMZ not ZMZ

3) line 203 "Bialutamide"

4) line 389 "bicalutamid"

5) line 429 "4oC"

Author Response

The reviewer comments: this study by Zalcman et al is a continuation of their previous studies on the role of AR in Glioblastoma and on the development of new antagonists aimed at inhibiting AR-induced intracellular signaling pathway in GBM.

The study is well-conducted and well-described, in my opinion, the manuscript can be accepted in this way but I suggest discussing the possibility of drug resistance due to the presence of alternative splicing forms of AR (see AR-V7 the authors previously described in GBM)

Our response: We thank the reviewer for acknowledging that our study is well-conducted and well-described and for his effort to improve our manuscript. In response to his suggestion to discuss the possibility of drug resistance due to the presence of alternative splicing forms of AR, we have added the following to the discussion section:

"It is important to acknowledge our prior publication, where we identified the expression of the AR variant (AR-V7) in ZH-161, potentially linked to androgen antagonists resistance in prostate cancer [3]. The juxtaposition of this finding with our earlier revelation of only a moderate response of ZH-161 to androgen receptor antagonists in in-vitro studies [3] introduces an intriguing aspect. The efficacy of Bic-Sol in the in-vivo ZH-161 model prompts consideration of potential explanations for this apparent discrepancy. Firstly, the resistance observed in prostate cancer may not seamlessly translate to glioblastomas, where distinct factors could influence the drug's efficacy, potentially accounting for the observed sensitivity in the ZH-161 model. Additionally, despite the expression of AR-V7, the ZH-161 line co-expresses AR-FL, adding an extra layer of complexity to the observed responses. The concurrent presence of AR-FL might contribute to the efficacy gained by the drug. This observation could strengthen our results. If the drugs exhibit efficacy in a cell line with AR-V7, coupled with the fact that only 30% of glioblastoma patients have AR-V7, it raises the intriguing possibility that these drugs might exert even greater efficacy in patients lacking this specific transition. While we acknowledge that these assertions are drawn from our observations and represent assumptions, future investigations, supported by rigorous clinical studies, will be imperative to validate and build upon these preliminary findings, deepening our understanding of the intricate interplay between AR-V7, AR-FL, and drug efficacy in the context of glioblastoma". (page 10 lines 285-304)

Comments on the Quality of English Language: English language is fine and fluent.

Reviewer comment: minor editing is required:

Our response: We have addressed the minor editing requirements by correcting the typos as suggested:

  1. line 99: "bic-sol" or "sol-bic"? This typo was corrected throughout the text on lines: 25,28,30,31,32,105,390.

2) line 201 TMZ not ZMZ this typo was corrected on line 207

3) line 203 "Bialutamide" this typo was corrected on line 209

4) line 389 "bicalutamid" this typo was corrected on line 429

5) line 429 "4oC" this typo was corrected on line 460

Reviewer 2 Report

Comments and Suggestions for Authors

Zalcman et al are bringing arguments for the therapeutic implications of AR expression in GBM and are making a step forward towards the possibility of using the AR antagonists in clinical practice.

As the solubility of AR antagonists is a practical issue in the clinical setting, I congratulate the team for establishing a soluble formula of bicalutamide that enables a better blood brain barrier passage and lower Tmax. However, although the PK study results are encouraging when performed in CD1 male mice, each strain of mice may present with different particularities, so a similar PK study should be performed in both male and female athymic nude mice, ie the strain that was used for the efficacy studies.

Overall, the use of the in vivo models seems appropriate, using both U87 and ZH-116 orthotopic xenografts in athymic nude-mice. However, although the control groups seem solid (12 to 20 mice depending on the studies), the treated samples are quite low 6-10 mice/group. I am actually surprised to see the statistical significance attained with these small mice samples.

More specifically, the Xtandi study in U87 (Figure 1) appears to present data pooled from two experiments or from one experiment depending on the condition. This shouldn’t be done as tumor growth can vary depending on cell batches. The two studies should be shown separately.

In the U87 model Figure 5:

1.       Although the difference in median survival between the controls and the mice treated with bicalutamide-sol is important, the difference between bicalutamide-sol-tmz combo and tmz alone mice is minimal (median OS 33 and 29 days respectively).

2.       Similarly, the combination with bicalutamide-sol (8mg/kg)-tmz seems to provide a modest survival advantage over the use of bicalutamide-sol alone (8mg/kg) (median OS 33 and 28 days respectively)

3.     Additionally, the group treated with bicalutamide-sol alone (4mg/kg) and the one with bicalutamide-sol alone (8mg/kg) have equal median OS of 28 days ie no dose-response effect is observed between these two doses

How would you comment on that?

The statistical p-values appear quite small for such a small effect and low (and heterogeneous) n numbers. Did the author perform an ANOVA test to compare the groups? This is how groups should be compared here.

Regarding the ZH-161 model figure 6:

-       As you previously showed, the ZH-161 express high levels of AR-V7 which is one of the main causes of bicalutamide resistance in prostate cancer. How would you explain the bicalutamide sensitivity of the ZH-161 model in this condition?

-       Although the ZH-161 in vivo model shows encouraging results, the differences between mice samples are extremely important 4 mice/group in bicalutamide treated group versus 12 mice in the control group.

I recommend these results to be confirmed in larger mice samples with more comparable n numbers.

Additional comments:

-no studies are presented to demonstrate that the observed therapeutic effects of enzalutamide and bicalutamide are really due to AR inhibition. This should be done at least in vitro, ideally in vivo

-It is not clear whether the AR pathway is active in U87 and ZH-161 cell lines

- there is no precision on the mice sex in the treatment studies: male or females? Although we don’t see any differences in AR expression between male and female GBM tissue, as the ligands (testosterone and DHT) are at different levels, this might be an issue that could play a role in treatment sensitivity

Comments on the Quality of English Language

The English terminology needs minor revisions

Author Response

Zalcman et al are bringing arguments for the therapeutic implications of AR expression in GBM and are making a step forward towards the possibility of using the AR antagonists in clinical practice.

The reviewer comment: As the solubility of AR antagonists is a practical issue in the clinical setting, I congratulate the team for establishing a soluble formula of bicalutamide that enables a better blood-brain barrier passage and lower Tmax. However, although the PK study results are encouraging when performed in CD1 male mice, each strain of mice may present with different particularities, so a similar PK study should be performed in both male and female athymic nude mice, i.e., the strain that was used for the efficacy studies.

Our response: We opted to investigate the pharmacokinetics (PK) in CD1 mice rather than Nude mice due to the intriguing aspect of their immune deficiency. We employed Nude mice to implant human glioblastoma in our study, aiming to prevent the rejection of the human glioblastoma. However, for the pharmacokinetic (PK) study, we chose CD1 mice instead of Nude mice. This decision was made to more closely replicate the conditions of patients with a preserved immune system, as the intended recipients of this drug have intact immune systems.

The reviewer comment: Overall, the use of the in vivo models seems appropriate, using both U87 and ZH-116 orthotopic xenografts in athymic nude-mice. However, although the control groups seem solid (12 to 20 mice depending on the studies), the treated samples are quite low 6-10 mice/group. I am actually surprised to see the statistical significance attained with these small mice samples. More specifically, the Xtandi study in U87 (Figure 1) appears to present data pooled from two experiments or from one experiment depending on the condition. This shouldn’t be done as tumor growth can vary depending on cell batches. The two studies should be shown separately.

Our response: In addressing your concern regarding the Xtandi study in U87 (Figure 1), we appreciate your observation. Initially, we conducted a pilot experiment involving a control group and a group treated with 20mg/kg Xtandi. Encouraged by positive results, we subsequently initiated an additional study comprising three study groups: control, 20mg/kg Xtandi, and 50mg/kg Xtandi. Given the absence of discernible differences between the control groups or the 20mg/kg groups in the two studies, we opted to combine them. This decision was made with the intent of potentially strengthening the results through consolidation, thereby providing a more comprehensive and robust representation of the experimental outcomes. We acknowledge the importance of transparency in reporting our methodologies, and in response to your observation, we have added a comment in the results section (page 2 line 83-89) to provide additional clarity and context.

In the U87 model Figure 5:

How would you comment on that?

The reviewer comment: Although the difference in median survival between the controls and the mice treated with bicalutamide-sol is important, the difference between bicalutamide-sol-tmz combo and tmz alone mice is minimal (median OS 33 and 29 days respectively).

Our response: Thank you for acknowledging the importance of the difference in median survival between the control group and the mice treated with bicalutamide-sol. Regarding your concern about the perceived minimal difference between bicalutamide-sol-tmz combo and TMZ alone mice, we would like to clarify that the actual differences are higher. In the TMZ alone group, the median survival is 26 days, while in the combo group, it is 33 days. Importantly, this difference is statistically significant with a p-value of 0.0184.

The reviewer comment: Similarly, the combination with bicalutamide-sol (8mg/kg)-tmz seems to provide a modest survival advantage over the use of bicalutamide-sol alone (8mg/kg) (median OS 33 and 28 days respectively)

Our response: We value your insightful observation. In response to your comment, we recognize that a potential resolution could involve refining the dose of the two drugs in future studies. This adjustment aims to explore and identify an optimal dose for the combination therapy, with the goal of maximizing efficacy and improving overall survival outcomes.

The reviewer comment: Additionally, the group treated with bicalutamide-sol alone (4mg/kg) and the one with bicalutamide-sol alone (8mg/kg) have equal median OS of 28 days, i.e., no dose-response effect is observed between these two doses

Our response: Regarding your concern about the observed equal median overall survival (OS) of 28 days in the group treated with bicalutamide-sol alone (4mg/kg) and the one with bicalutamide-sol alone (8mg/kg), several factors may contribute to this lack of a dose-response effect. It's important to note that the two doses were not administered in the same experiments, which could introduce variability. Additionally, the similarity in median OS might be influenced by potential cell line batch effects. Furthermore, the absence of a dose-response effect might suggest that doubling the amount of the drug is not sufficient to elicit a significant difference in overall survival outcomes. Future investigations will consider these factors, and we appreciate your consideration and insights in interpreting these results.

The reviewer comment: The statistical p-values appear quite small for such a small effect and low (and heterogeneous) n numbers. Did the author perform an ANOVA test to compare the groups? This is how groups should be compared here.

Our response: We are grateful for your comments related to the statistical analysis in our study. Regarding the statistical p-values, it is noteworthy that most p-values fall below the conventional significance threshold of 0.05, ranging between 0.0177 to 0.00364. This suggests that the observed differences between the study groups are statistically significant. While we did not conduct an ANOVA test in this context, our rationale for choosing Kaplan-Meier survival analysis stems from the nature of our time-to-event data and the presence of censored observations. This method allows for a robust exploration of survival probabilities over time, particularly suitable for the nuances of our dataset.

Regarding the ZH-161 model figure 6:

The reviewer comment: As you previously showed, the ZH-161 express high levels of AR-V7 which is one of the main causes of bicalutamide resistance in prostate cancer. How would you explain the bicalutamide sensitivity of the ZH-161 model in this condition?

Our response: your concern regarding the high expression of AR-V7 in ZH-161, potentially contributing to bicalutamide resistance in prostate cancer, is justified, especially when considering our previous publication that indicated only a moderate response of ZH-161 to androgen receptor antagonists in in-vitro studies. Several potential explanations may clarify this discrepancy: the resistance initially observed in prostate cancer may not necessarily translate to glioblastomas, where other factors could influence drug efficacy, and the observed sensitivity in the ZH-161 model might be attributed to these variations. Additionally, despite the expression of AR-V7, the ZH-161 line co-expressed AR-FL. The presence of AR-FL might contribute to the efficacy gained by the drug.

This observation could strengthen our results. If the drugs demonstrate efficacy in a cell line with AR-V7, and considering that, only 30% of glioblastoma patients have AR-V7, it raises the possibility that these drugs might be even more effective in patients without this particular transition.

While we acknowledge that these are assumptions drawn from our observations, we believe they offer plausible explanations for the observed responses. Further investigations and clinical studies will be crucial to validate and expand upon these findings. (This was discussed at page 10 lines 285-304)

The reviewer comment: Although the ZH-161 in vivo model shows encouraging results, the differences between mice samples are extremely important 4 mice/group in bicalutamide treated group versus 12 mice in the control group.

Our response: Thank you for recognizing the encouraging results observed in the ZH-161 in vivo model. We acknowledge the importance of addressing the differences in the number of mice samples across groups, particularly with four mice per group in the 4mg/kg Sol-bic-treated group versus 12 mice in the control group. While there are substantial differences between the study groups, it is essential to highlight that in the two most critical groups, the control and the 8mg/kg Sol-bic-treated group groups, there are 12 and 10 mice per group, respectively. This robust sample size in these key groups substantiates the main point regarding the efficacy of sol bicalutamide. Moreover, even in the groups with a smaller number of animals, the observed differences between the groups remain statistically significant. This underscores the statistical validity of the results, contributing to the overall strength of the findings.

The reviewer comment: I recommend these results to be confirmed in larger mice samples with more comparable n numbers.

Our response: We concur with the reviewer's recommendation. While we acknowledge that such experiments would likely enhance the robustness of the study, unfortunately, performing these additional experiments is not feasible within the time scope of this revision.

Additional comments:

The reviewer comments: no studies are presented to demonstrate that the observed therapeutic effects of enzalutamide and bicalutamide are really due to AR inhibition. This should be done at least in vitro, ideally in vivo - It is not clear whether the AR pathway is active in U87 and ZH-161 cell lines

Our response: These two issues were addressed in our previous publications: Zalcman N, et al., Oncotarget. 2018 Apr 13;9(28):19980-19993 and Zalcman N, et al., Int J Mol Sci. 2021 Oct 11;22(20).

The reviewer comment: there is no precision on the mice sex in the treatment studies: male or females? Although we don’t see any differences in AR expression between male and female GBM tissue, as the ligands (testosterone and DHT) are at different levels, this might be an issue that could play a role in treatment sensitivity

Our response: Thank you for your valuable comment. The sex of the mice (male) has been included in the Material and Methods section (page 12, line 380).

Round 2

Reviewer 2 Report

Comments and Suggestions for Authors

Although the differences in n number between treatment groups are still a methodological problem to me, at least the n numbers are clearly indicated in the graphs which makes this information obvious to the reader. I still insist however that the two Xtandi studies be presented in two separate graphs and cannot be pooled together, because, as the authors point out in their response to reviews, different experiments carry variability due mice lots and cell line batch effects. 

Author Response

the reviewer's comment: Although the differences in n number between treatment groups are still a methodological problem to me, at least the n numbers are clearly indicated in the graphs which makes this information obvious to the reader. I still insist however that the two Xtandi studies be presented in two separate graphs and cannot be pooled together, because, as the authors point out in their response to reviews, different experiments carry variability due mice lots and cell line batch effects. 

Our response: We appreciate the reviewer's thorough examination of our manuscript. In response to the reviewer's suggestion, we have plotted the two Xtandi studies separately, presenting them in two distinct graphs within Supplementary Figure 1. This adjustment allows for a clearer representation of the data while addressing concerns related to potential variability arising from different experiments. We have explicitly referenced this supplementary figure in the Results section (lines 83-91) to ensure that readers are directed to the relevant information.